# A Novel *FGFR1* Missense Mutation in a Portuguese Family with Congenital Hypogonadotropic Hypogonadism

**DOI:** 10.3390/ijms23084423

**Published:** 2022-04-17

**Authors:** Lúcia Fadiga, Mariana Lavrador, Nuno Vicente, Luísa Barros, Catarina I. Gonçalves, Asma Al-Naama, Luis R. Saraiva, Manuel C. Lemos

**Affiliations:** 1Serviço de Endocrinologia, Diabetes e Metabolismo, Centro Hospitalar Universitário de Coimbra, 3000-075 Coimbra, Portugal; lmsfadiga@gmail.com (L.F.); mariana.lavrador@hotmail.com (M.L.); nunovic@gmail.com (N.V.); mlmcbarros@gmail.com (L.B.); 2CICS-UBI, Health Sciences Research Centre, University of Beira Interior, 6200-506 Covilhã, Portugal; cigoncalves@fcsaude.ubi.pt; 3Sidra Medicine, Doha P.O. Box 26999, Qatar; aalnaama2@sidra.org

**Keywords:** hypogonadotropic hypogonadism, Kallmann syndrome, FGFR1, fibroblast growth factor receptor 1, genetics, mutation

## Abstract

Congenital hypogonadotropic hypogonadism (CHH) is a rare reproductive endocrine disorder characterized by complete or partial failure of pubertal development and infertility due to deficiency of the gonadotropin-releasing hormone (GnRH). CHH has a significant clinical heterogeneity and can be caused by mutations in over 30 genes. The aim of this study was to investigate the genetic defect in two siblings with CHH. A woman with CHH associated with anosmia and her brother with normosmic CHH were investigated by whole exome sequencing. The genetic studies revealed a novel heterozygous missense mutation in the Fibroblast Growth Factor Receptor 1 (*FGFR1)* gene (NM_023110.3: c.242T>C, p.Ile81Thr) in the affected siblings and in their unaffected father. The mutation affected a conserved amino acid within the first Ig-like domain (D1) of the protein, was predicted to be pathogenic by structure and sequence-based prediction methods, and was absent in ethnically matched controls. These were consistent with a critical role for the identified missense mutation in the activity of the FGFR1 protein. In conclusion, our identification of a novel missense mutation of the *FGFR1* gene associated with a variable expression and incomplete penetrance of CHH extends the known mutational spectrum of this gene and may contribute to the understanding of the pathogenesis of CHH.

## 1. Introduction

Congenital hypogonadotropic hypogonadism (CHH) is a rare condition characterized by absent or incomplete puberty and infertility due to a deficient production, secretion or action of the gonadotropin-releasing hormone (GnRH) [1,2]. Patients have low serum concentrations of the gonadotropins LH (luteinizing hormone) and FSH (follicle-stimulating hormone) and low concentrations of sex steroids [1,2].

CHH has a significant clinical heterogeneity, and the diagnosis is often difficult to differentiate from constitutional delay of growth and puberty [3]. CHH includes Kallmann Syndrome, which is characterized by GnRH deficiency with a defective sense of smell (i.e., anosmia or hyposmia), and CHH without olfactory defects (normosmic CHH). Non-reproductive phenotypes may also be found in CHH patients, such as midline facial and brain defects, dental agenesis, sensorineural hearing impairment, renal agenesis and skeletal defects [1,2].

There are over 30 genes associated with CHH [4,5]. Although most families reveal a Mendelian inheritance pattern (X-linked, autosomal recessive or autosomal dominant), oligogenicity may occur in up to 20% of cases [6]. Furthermore, family studies often show evidence of incomplete penetrance and variable expressivity, including milder phenotypes such as delayed puberty or isolated anosmia [4,5]. Despite the large number of CHH-associated genes, about 50% of CHH patients remain without identification of a genetic aetiology [4,5].

In this study, we present the clinical and genetic studies of two siblings affected with Kallmann Syndrome and normosmic CHH, respectively, leading to the identification of a novel missense mutation in the Fibroblast Growth Factor Receptor 1 (*FGFR1*) gene.

## 2. Results

### 2.1. Clinical Studies

Patient 1. A 19 year old woman was observed due to primary amenorrhoea and the absence of secondary sexual characteristics. She also mentioned anosmia. She had an otherwise normal development, had no relevant medical history, and was not taking any medication. Her parents were not consanguineous and had no relevant past history, except for her mother who had a late spontaneous menarche, at the age of 18. She had two brothers (21 and 17 years old, respectively) with no reported health or development problems at that time. Her physical examination showed a good general condition and no dysmorphism. Her height and weight were 167 cm and 52 kg, respectively (body mass index (BMI) 18.6 kg/m^2^). Pubertal development corresponded to Tanner stage 1. Blood tests revealed estradiol 18 pg/mL (normal range (NR) 20–60), FSH 0.3 mIU/mL (NR 1.4–9.6), LH <0.1 mIU/mL (NR 0.8–26.0), thus indicating a diagnosis of hypogonadotropic hypogonadism. No other hormonal or blood chemistry abnormalities were detected. A GnRH stimulation test was performed, and the levels of FSH and LH increased to 3.9 mIU/mL and 1.7 mIU/mL, respectively. Her karyotype was normal. A pelvic ultrasound revealed a small prepubertal uterus (4.8 × 2.6 × 2.7 cm) and small left and right ovaries (1.7 cm and 1.5 cm, respectively). Lumbar spine densitometry showed a Z-score of –3.42. A head magnetic resonance imaging (MRI) revealed a normal pituitary gland and olfactory bulbs, without any midline defects. She started 17 β-estradiol therapy with a transdermic long-action formulation. She showed a good response, with adequate pubertal development. She had a breakthrough bleeding after a year and then started combined estroprogestin therapy. At the age of 21, she weighed 62 kg and was 171 cm tall, and had a complete pubertal development. At the age of 23, lumbar spine densitometry revealed an increase in bone mass, with a Z-score of –2.1. At the age of 29, she underwent gonadotropin treatment for fertility. After a second cycle of follicular stimulation, it was possible to collect oocytes for in vitro fertilization and to generate embryos that were successfully transferred. Pregnancy was uneventful, and she gave birth to a full-term healthy girl weighing 3.830 kg. She was last evaluated at the age of 37 and remained under oral estroprogestin therapy, with good physical and mental health, and was planning a second pregnancy.

Patient 2. A 30 year old male was referred by his general practitioner due to the absence of pubertal development. He was the younger brother of Patient 1. He complained about the absence of a beard, a short penis, erectile dysfunction and poor muscle strength. He reported a normal sense of smell. He had no relevant past history and was not taking any medication. At physical examination, he had a prepubertal phenotype, with no facial hair, a high-pitched voice, a micropenis, sparse pubic hair and testicular volumes of approximately 6–7 mL (Tanner stage 2). He was 167 cm tall and weighed 64 kg (BMI 22.9 kg/m^2^). Blood tests revealed total testosterone 0.5 ng/mL (NR 3.0–10.0), FSH 1.0 mIU/mL (NR 0.9–15.0), LH 0.4 mIU/mL (NR 1.3–13.0), thus indicating a diagnosis of hypogonadotropic hypogonadism. He had hypercholesterolemia. No other hormonal or blood chemistry abnormalities were detected. Lumbar spine densitometry showed a Z-score of –4.5. A head MRI revealed an olfactory groove with a normal configuration but with poor depth (Keros type I), normal olfactory bulbs and tracts, and a normal pituitary gland. He started intramuscular testosterone enanthate 125 mg every 3 weeks, with a good response. After 10 months, he showed a deepening of voice, the appearance of a beard, and an increase in penile size and pubic hair (Tanner stage 5). Testicular volumes were approximately 8–10 mL. The patient also referred an improvement in general wellbeing, muscle strength, sexual performance and the appearance of ejaculation. He was last evaluated at the age of 35 and remained well under testosterone enanthate 250 mg every 4 weeks and had no plans for parenthood.

### 2.2. Genetic Studies

Whole exome and Sanger sequencing identified a heterozygous missense variant in exon 3 of the *FGFR1* gene (NM_023110.3: c.242T>C, p.Ile81Thr) in both siblings. The variant was absent in the Genome Aggregation Database (gnomAD) [7] and in a panel of 200 control Portuguese individuals. The screening of the remaining family members showed that the variant was present in the father and absent in the mother and unaffected brother (Figure 1a,b). The variant had high scores for a damaging effect on Polymorphism Phenotyping v2 (PolyPhen-2) [8] (score 0.98, on a scale from 0 to 1), Combined Annotation-Dependent Depletion (CADD) [9] (score 24, on a scale from 0 to 30+), Genomic Evolutionary Rate Profiling (GERP) [10] (score 5.58, on a scale from –12 to 6.17), and Mutation Taster-2 [11] (score 0.99, on a scale from 0 to 1). The variant affected an amino acid (isoleucine, Ile) that showed a high degree of conservation across species (phyloP score 3.209, on a scale from –14 to +6; and phastCons score 1, on a scale from 0 to 1) [11] (Figure 1c). The amino acid substitution was predicted to reduce the stability of the protein (ΔΔG: –0.575 kcal/mol) [12] (Figure 1d). The variant fulfilled the American College of Medical Genetics and Genomics (ACMG) criteria [13] for “Likely Pathogenic” (criteria PM1, PM2, PP2, PP3). No other non-synonymous rare variants were identified in the 102 analyzed genes [14].

## 3. Discussion

Our study of two siblings with CHH identified a novel *FGFR1* missense mutation. The FGFR1 protein is a transmembrane receptor that comprises an extracellular region of three immunoglobulin (Ig)-like domains (D1, D2, and D3), a transmembrane helix, and a cytoplasmic tyrosine kinase domain [15]. This receptor plays an important role in the development of GnRH neurons, as well as in their migration from the nasal placode [16]. Mutations in *FGFR1* (previously named *KAL2*) were first associated with Kallmann syndrome in 2003 [17]. Since then, at least 254 mutations have been linked to CHH, in patients with and without anosmia [18]. *FGFR1* is one of the most commonly implicated genes in CHH [4,5]. In a previous Portuguese multicentric study, *FGFR1* mutations were identified in 12% of CHH patients, representing the second most common cause of CHH, after Chromodomain Helicase DNA Binding Protein 7 (*CHD7*) mutations [19,20,21,22].

The identified mutation leads to the substitution of an isoleucine by a threonine at amino acid position 81, located within the first Ig-like domain (D1) of the protein, which spans amino acids 25 to 119. A total of 21 other missense mutations have been reported in the D1 domain, but so far, none have been identified at amino acid position 81 [18]. The mutation is predicted to be pathogenic by sequence- and structure-based prediction methods [8,9,10,11,12]. Furthermore, the high degree of conservation of the mutated amino acid across species and the absence of the variant in ethnically matched controls are consistent with a critical role for the identified missense mutation in the activity of the FGFR1 protein. However, we did not perform in vitro functional studies that could potentially confirm a defect in the production, localization, or function of the mutated protein, and provide further support for the pathogenicity of the *FGFR1* variant.

The phenotype associated with the *FGFR1* mutation varied among the family members. The affected sister had anosmia, but the affected brother had a normal olfactory function. However, olfaction was self-assessed by the patients, and no formal testing was performed that could objectively quantify this trait. The mutation was inherited from the father who had no relevant medical history, therefore indicating a case of incomplete penetrance. Variable expression and incomplete penetrance are commonly observed within families carrying *FGFR1* mutations [23]. Phenotypes can vary from CHH to delayed puberty to normal reproductive function, with or without anosmia and other non-reproductive abnormalities [23]. This suggests a role for other genes that modify the phenotype of *FGFR1* mutations. Indeed, oligogenic inheritance has been identified in several families and sometimes explains the variable expression of *FGFR1* mutations [19,24]. However, our analysis of an extended panel of 102 genes implicated in CHH [14] did not reveal any other genetic variants that could account for oligogenicity in this family.

With appropriate hormone replacement therapy, patients with CHH can develop secondary sexual characteristics, and maintain normal sex hormone levels and a healthy sexual life [1]. The deficiency of GnRH in CHH causes infertility in men and women due to failed gamete production and/or maturation. However, CHH is a treatable form of infertility, as in most cases, spermatogenesis and ovulation can be induced by pulsatile GnRH or exogenous gonadotropin administration [1]. Our female patient was successfully treated for her infertility, but because of the autosomal dominant mode of inheritance of *FGFR1* mutations, there remains a 50% chance of passing on the gene defect to her offspring.

In conclusion, our identification of a novel missense mutation of the *FGFR1* gene, associated with the variable expression and incomplete penetrance of CHH, extends the known mutational spectrum of this gene and may contribute to the understanding of the pathogenesis of CHH.

## 4. Materials and Methods

The genetic studies were approved by the Institutional Ethics Committee of the Faculty of Health Sciences-University of Beira Interior, Portugal (Ref: CE-FCS-2012-012), and by the Institutional Review Board (IRB) for the protection of human subjects in Sidra Medicine, Qatar (IRB Ref: 157003). Written informed consent was obtained from all subjects.

Genomic deoxyribonucleic acid (DNA) was extracted from peripheral blood leucocytes of the patients and unaffected family members. DNA from Patient 2 was used for whole exome sequencing (WES), followed by the analysis of a virtual gene panel consisting of 102 genes that have been associated with CHH or suggested to be candidate genes for CHH [14]. For WES, targeted enrichment was performed using Agilent SureSelectXT All Exon V6 (Agilent Technologies, Santa Clara, CA, USA), and the target regions were sequenced on the Illumina HiSeq 4000 platform (Illumina, Inc., San Diego, CA, USA) with paired-end reads of 150 bp and 100 × raw read coverage. Reads were mapped to the human reference genome GRCh37/hg19 (hs37d5) using the Burrows–Wheeler Aligner (BWA-MEM) software [25]. Variants were called using SAMtools (v.1.6) [26] and annotated using the SnpEff (v.4.3) [27] and dbNSFP (v3.0) [28] tools. Data files were generated using the Genome Analysis Toolkit v3.5 (GATK) [29] best practices workflow.

Variants in the genes of interest were selected if they were: (1) located in coding exons or adjacent splice sites, (2) non-synonymous, (3) absent or rare (population frequency <0.1%) in the gnomAD [7], and (4) absent in an in-house database of 200 control Portuguese individuals.

Filtered variants were confirmed by conventional Sanger sequencing using a CEQ DTCS sequencing kit (Beckman Coulter, Fullerton, CA, USA) and an automated capillary DNA sequencer (GenomeLab TM GeXP, Genetic Analysis System, Beckman Coulter, Fullerton, CA, USA), and were screened for in the other family members.

The functional consequences of the *FGFR1* variant were predicted using the PolyPhen-2 [8], CADD [9], GERP [10], and Mutation Taster-2 [11] tools. The impact of the *FGFR1* variant on protein conformation and stability was analyzed with the DynaMut tool [12] and visualized with PyMOL (PyMOL Molecular Graphics System, Version 2.4.1 Schrödinger, LLC, New York, NY, USA). The *FGFR1* variant was analyzed by VarSome [30] and classified according to ACMG criteria [13].

The nomenclature of the variant was based on the *FGFR1* cDNA reference sequence (GenBank accession number NM_023110.3).

## Figures and Tables

**Figure 1 ijms-23-04423-f001:**
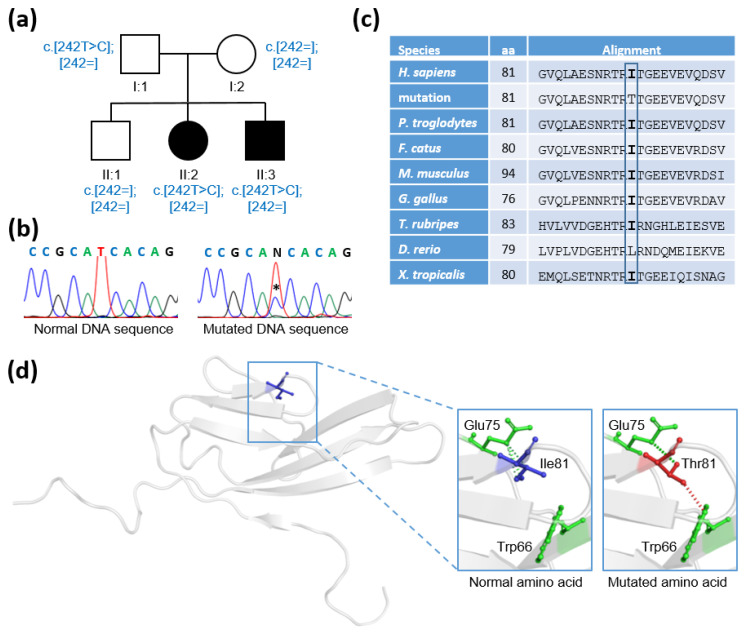
(**a**) Pedigree of the patients with congenital hypogonadotropic hypogonadism (CHH). Individuals are represented as males (squares), females (circles), affected (filled symbol), and unaffected (open symbol). The genotype is shown for each family member (the mutated allele is represented as c.[242T>C] and the normal allele is represented as c.[242=]). (**b**) Partial DNA sequence of exon 3 of the Fibroblast Growth Factor Receptor 1 (*FGFR1*) gene showing a heterozygous missense mutation (NM_023110.3: c.242T>C, p.Ile81Thr) (asterisk) which was present in the patients and in their unaffected father (a normal sequence is presented for comparison). (**c**) Alignment of the local amino acid sequence across different species showing a high degree of conservation of the amino acid isoleucine (I) in this protein domain. (**d**) Mapping of the p.Ile81Thr missense mutation onto the crystal structure of the FGFR1 immunoglobulin-like domain 1 (D1) (Protein Data Bank ID: 2CR3). The normal amino acid Ile81 (blue), the mutated amino acid Thr81 (red) and the amino acids in the vicinity of residue 81 (Glu75 and Trp66) (green) are highlighted (magnified inserts). Ile81 maintains three hydrophobic contacts with Glu75, whereas Thr81 maintains only one hydrophobic bond with Glu75 and a hydrogen bond with Trp66.

## Data Availability

The data that support the findings of this study are available from the corresponding author on reasonable request.

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
