# Peer review of "A Novel FGFR1 Missense Mutation in a Portuguese Family with Congenital Hypogonadotropic Hypogonadism"

_ijms, 2022, doi:10.3390/ijms23084423_

Round 1

Reviewer 1 Report

This work would be interesting if supported by at least  in vitro functional studies. Experiments demonstrating defects in the production/localization of the mutated protein are needed to conclude the possible pathogenicity of the FGFR1 variant. 

Author Response

Reviewer #1.

This work would be interesting if supported by at least in vitro functional studies. Experiments demonstrating defects in the production/localization of the mutated protein are needed to conclude the possible pathogenicity of the FGFR1 variant.

>>>Authors’ response: We would like to thank the reviewer for this important comment. We agree that the mutation could be further studied by in vitro functional experiments to demonstrate the pathogenicity of the FGFR1 variant. Although we have supported our conclusions by bioinformatics analyses and by ACMG criteria (which are quite stringent indeed), we acknowledge that in vitro experiments could provide additional support and more conclusive evidence. However, such experiments are very time-consuming and expensive and do not always result in conclusive evidence. We regret to say that we are not in a position to be able to carry out such studies at present. We have included the lack of functional studies as a limitation in the discussion as follows: ”(…) However, we did not perform in vitro functional studies that could potentially confirm a defect in the production, localization, or function of the mutated protein, and provide further support for the pathogenicity of the FGFR1 variant”. We hope the reviewer finds this an acceptable compromise.

Reviewer 2 Report

Paper is presenting novel FGFR1 change that is causative for CHH in reported family.

Some minor criticism:

1. line 107: 200 normal Portuguese individuals - better is "200 control Portuguese individuals"

2. line 113: "Mutation Taster" is old and not in use anymore - I guess Mutation Taster2 was used or newest Mutation Taster2021

3. line 114: support "degree of conservation" with some real data or calculation - PhyloP / phastCons (Mutation Taster2) or MaxEntScan

4 . line127: Ile81Thr should be replaced with p.Ile81Thr

5. Figure 1a: The rules for pedigree drawing should be used -symbols description I:2, I:1, II:1, II:2, II:3. Also I encourage to use genotype description in regard of variant presence or absence

(+) replace for c.[242T>C];[242=] and (-) for c.242= . This information fulfil HGVS recommendation and is self-explanatory (zygosity)  

Author Response

Reviewer #2

Paper is presenting novel FGFR1 change that is causative for CHH in reported family.

Some minor criticism:

  1. line 107: 200 normal Portuguese individuals - better is "200 control Portuguese individuals"

>>>Authors’ response: We thank the reviewer for this suggestion. We have made the changes in the methods and results sections.

  1. line 113: "Mutation Taster" is old and not in use anymore - I guess Mutation Taster2 was used or newest Mutation Taster2021

>>>Authors’ response: Thank you for pointing this out. We repeated the analysis with Mutation Taster2 and obtained the same results. We have updated the reference for Mutation Taster2. We also tried Mutation Taster2021 but, for some unknown reason, it does not recognize the alignment of the amino acid sequence with chimpanzee and mouse in the species alignment. Therefore, we chose to present the alignment obtained through Mutation Taster2.

  1. line 114: support "degree of conservation" with some real data or calculation - PhyloP / phastCons (Mutation Taster2) or MaxEntScan

>>>Authors’ response: Thank you for this useful suggestion. We retrieved the PhyloP / phastCons in Mutation Taster2 and present the scores in the results section.

4 . line127: Ile81Thr should be replaced with p.Ile81Thr

>>>Authors’ response: We apologize for the mistake. We have corrected this.

  1. Figure 1a: The rules for pedigree drawing should be used -symbols description I:2, I:1, II:1, II:2, II:3. Also I encourage to use genotype description in regard of variant presence or absence

(+) replace for c.[242T>C];[242=] and (-) for c.242= . This information fulfil HGVS recommendation and is self-explanatory (zygosity) 

>>>Authors’ response: Thank you for drawing our attention to this. We have changed the figure and legend accordingly.

Round 2

Reviewer 1 Report

I thank authors for considering my comments by I think that the work provided is not sufficient for a publication in IJMS. The results sound very promising preliminary studies and surely I will take them in consideration for an improved future paper. However, in the present state I am sorry to reject this work.